# Heterogeneity and Functions of Tumor-Infiltrating Antibody Secreting Cells: Lessons from Breast, Ovarian, and Other Solid Cancers

**DOI:** 10.3390/cancers14194800

**Published:** 2022-09-30

**Authors:** Yasmine Lounici, Olivia Le Saux, Gabriel Chemin, Pauline Wajda, Sarah Barrin, Justine Berthet, Christophe Caux, Bertrand Dubois

**Affiliations:** 1Centre de Recherche en Cancérologie de Lyon, Inserm U1052, CNRS 5286, 69008 Lyon, France; 2Centre Léon Bérard, Université de Lyon, Université Claude Bernard Lyon 1, 69008 Lyon, France; 3Service D’Oncologie Médicale, Centre Léon Bérard, 69008 Lyon, France; 4Lyon Immunotherapy of Cancer Laboratory (LICL), Centre Léon Bérard, 69008 Lyon, France

**Keywords:** breast cancer, ovarian cancer, antibody-secreting cells, antibodies, heterogeneity

## Abstract

**Simple Summary:**

B cells are gaining increasing recognition as important contributors to the tumor microenvironment, influencing, positively or negatively, tumor growth, patient survival, and response to therapies. Antibody secreting cells (ASCs) constitute a variable fraction of tumor-infiltrating B cells in most solid tumors, and they produce tumor-specific antibodies that can drive distinct immune responses depending on their isotypes and specificities. In this review, we discuss the current knowledge of the heterogeneity of ASCs infiltrating solid tumors and how both their canonical and noncanonical functions shape antitumor immunity, with a special emphasis on breast and ovarian cancers.

**Abstract:**

Neglected for a long time in cancer, B cells and ASCs have recently emerged as critical actors in the tumor microenvironment, with important roles in shaping the antitumor immune response. ASCs indeed exert a major influence on tumor growth, patient survival, and response to therapies. The mechanisms underlying their pro- vs. anti-tumor roles are beginning to be elucidated, revealing the contributions of their secreted antibodies as well as of their emerging noncanonical functions. Here, concentrating mostly on ovarian and breast cancers, we summarize the current knowledge on the heterogeneity of tumor-infiltrating ASCs, we discuss their possible local or systemic origin in relation to their immunoglobulin repertoire, and we review the different mechanisms by which antibody (Ab) subclasses and isoforms differentially impact tumor cells and anti-tumor immunity. We also discuss the emerging roles of cytokines and other immune modulators produced by ASCs in cancer. Finally, we propose strategies to manipulate the tumor ASC compartment to improve cancer therapies.

## 1. Introduction

B cells and antibody secreting cells (ASC) are critical effectors of the adaptive immune system. Although their importance in infectious disease and vaccination is well-recognized, their roles in cancer and response/resistance to immunotherapies have been overlooked for a long time in favor of T cells. B cell infiltration has been documented early on in many solid tumors using conventional IHC with markers such as CD19 and CD20. B cells are rarely found alone and are usually closely associated with T cells and myeloid cells, and can account for up to 60% of the immune infiltrate in some patients [1,2,3,4,5]. The diversity of tumor-infiltrating (Ti) B cells can be captured using flow cytometry and medium- to high-dimension in situ tumor tissue staining by combining B cell subset-specific markers, computational analyses of bulk RNAseq data using gene signatures or deconvolution tools, single-cell RNAseq, and, more recently, spatial transcriptomics. Ti B cells can encompass naïve and activated B cells, several populations of antigen-experienced memory B cells and ASCs, and, occasionally, germinal center (GC) like B cells [1,6,7]. The presence of this continuum of B cell differentiation stages usually typifies so-called immune hot tumors where B cells are functionally organized with other immune cells in so-called tertiary lymphoid structures (TLS), resembling secondary lymphoid organs, with segregated B cell and T cell zones, mature dendritic cells (DC), and specialized blood vessels. Such structures can initiate and/or amplify powerful in situ cellular and antibody responses, and are associated with better patient prognosis and response to immune therapies [8]. In addition, B cells producing suppressive cytokines such as IL10, e.g., so-called regulatory B cells (Bregs), were also identified in the tumor microenvironment (TME) in certain tumor types and were demonstrated to dampen anti-tumor immunity [9,10,11,12,13,14,15,16]. Ti-ASCs consist of plasmablasts and plasma cells, and constitute a multifunctional B cell subset that can impact tumor cells and cells from the TME in multiple ways. Indeed, each ASC can not only produce a huge quantity of monoclonal immunoglobulins of a unique isotype that recognize a specific (tumor) antigen (Ag), but can also exert noncanonical functions through the release of pro- or anti-inflammatory cytokines and/or the expression of immune checkpoint ligands able to modulate antitumor immunity. In this review, we will highlight, mainly in breast and ovarian cancers (BC and OC), the phenotypic and functional diversities of Ti-ASCs and how both their Ab-dependent and -independent functions shape antitumor immunity and impact patient prognosis.

## 2. Humoral Response Development and Prognosis Impact in Breast and Ovarian Cancers

### 2.1. ASCs Infiltrate Breast and Ovarian Tumors

Antibody-secreting cells, identified using morphological features or by expression of CD38 or CD138 through immunohistochemistry or immunofluorescence, can be detected in approximately 30% of BC and high-grade serous OC (range from 12% to 70% according to studies) and their density within the tumor infiltrate can vary widely from patient to patient [3,7,17,18,19]. Such variations can be explained by several factors, including (i) the type of analyzed tissue (e.g., ASCs were found to be increased in ovaries and metastases compared to fallopian tubes [20]), (ii) the histology and molecular classification of the tumor (e.g., ASC infiltration is typically higher in medullary [17,19], HER2+ and triple-negative BC compared to ER+ BC [21]), (iii) tumor-specific characteristics, (e.g., expression of tumor-associated Ags as described in OC where ASC infiltration was positively associated with the expression of the cancer–testis antigens NY-ESO-1, MAGEA1 and CTAG2 [2]), and iv) lack of a truly specific ASC marker (CD138, often used as a biomarker to identify ASC, can be expressed by other cells and is poorly detectable on immature ASC, e.g., plasmablasts (PBs), which infiltrate up to 80% of tumors [18,22]). However, the degree of ASC infiltration does not seem to vary with tumor mutation load, *BRCA1/2* status and P53 mutations [7]. Several general features are associated with Ti-ASCs. They preferentially localize in the tumor stroma [23,24] in the vicinity of CD8^+^ and CD4^+^ T cells, B cells, and TLSs [7,17,25]. In most cancer types, Ti-ASCs produce mostly IgG antibodies (Abs), although this may vary depending on the type of tissue (nonmucosal vs. mucosal origin of the cancer tissue) [7,19,26,27,28,29,30,31,32,33,34]. Recently, Biswas et al., reported intriguing results showing that ASCs infiltrating OC were mainly producing IgAs [35], contrasting with other studies indicating a dominance of IgG-expressing cells [7,26]. The reasons for these discrepancies remain currently unknown but could be linked to the use of tissue from different anatomical locations, to different treatments received by patients, and/or to the possible presence of adjacent normal mucosal tissue.

### 2.2. Antibodies in Breast and Ovarian Cancer Patients

Antibodies directed against a broad array of tumor- and self-Ag are frequently detected in the serum and the tumor microenvironments (TME) of cancer patients. These Ags include aberrantly- and over-expressed proteins, oncoviral and intracellular proteins, endogenous retroelements, and occasionally neoantigens derived from tumor mutations. Early approaches to studying the repertoire of antibodies in cancer patients include serological expression cloning (SEREX) [36] and phage display [37] methods. For example, NY-ESO-1, a well-known cancer testis antigen aberrantly expressed in various solid tumors, including BC and OC, was discovered by SEREX based on its capacity to induce potent humoral response in cancer patients [38]. The magnitude of such antibody responses varies greatly according to the cancer type and from patient to patient. Profiling of serum antibody specificities using microarrays assembling more than 8000 proteins has indeed revealed that more than 200 Ags were targeted by IgGs in OC, compared to less than 30 in pancreatic cancers, revealing their difference of immunogenicity [39]. Levels of circulating antibodies to tumor and self-Ags can have some prognostic and diagnostic value. For instance, antibodies against mucin 1 (MUC1) can be detected in the serums of patients at early stages of BC and OC and serve as a marker of good prognosis [40,41], whereas anti-p53 antibodies have been associated with an unfavorable outcome in BC [42,43]. Few studies so far have investigated other antibody isotypes than IgG. Interestingly, anticalreticulin IgGs and IgAs were reported in various solid cancers, and IgAs were more often associated with breast tumor metastasis than were IgGs [44], suggesting differential roles of the two Ab isotypes. In BC, Ab levels to different sets of Ag have been proposed for early detection of the disease, including the in situ stage [45]. Such diagnostic value of circulating Ab is best exemplified in paraneoplastic neurological diseases, which are rare autoimmune diseases that develop in a fraction of cancer patients. Elevated levels of Ab to so-called onconeural-Ag, e.g., Ags expressed physiologically by neuronal cells and aberrantly by tumor cells, are not only used for the diagnosis of the neurological disease but can also predict the type and tissue location of tumors, which can be indolent, causing it to be difficult to diagnose in a significant proportion of patients [46,47,48].

Circulating tumor-specific antibodies likely originate from plasma cells (PCs) residing in classical niches, such as the bone marrow and spleen, but also from PCs present in the TME. Profiling of supernatants of B cells isolated from lung tumors and activated in vitro and of supernatants obtained during dissociation of breast tumors with focused sets of Ag, revealed IgGs and IgAs against one or several tumor Ag in each patient, indicating local production of tumor-specific Ab in the TME [49,50]. Interestingly, while most antibody reactivities were detected in both the serum and tumor, some were restricted to a single compartment, indicating that systemic and local Ab responses in cancer patients are partly disconnected. It is indeed likely that the repertoire of Ab in the TME is more restricted and focused on tumor Ag, especially in tumors with TLS, which promote the Ag-driven expansion of B cells and their differentiation into PCs [51,52].

### 2.3. Origin of Tumor-Infiltrating ASCs

There is increasing evidence that Ti-ASCs consist not only of cells differentiated outside the TME and attracted from blood to the tumor bed, but also of cells that have locally differentiated from recruited naive and/or memory B cells. It is now well established that PBs generated in secondary lymphoid organs (SLO) can acquire different chemokine receptors, like CXCR4, CCR9, and CCR10, allowing their migration to the bone marrow to constitute a contingent of long-lived PCs and to different effector sites including mucosal tissues. Studies in vaccination models revealed that PBs can also express CXCR3 allowing their migration to inflammatory sites in response to gradients of the inflammatory chemokines CXCL9, CXCL10, or CXCL11 [53]. In OC, Kroeger et al., indeed showed that Ti-IgG^+^ ASC universally expressed CXCR3, suggesting that this chemokine receptor may contribute to their recruitment [7].

With the discovery of TLSs in certain tumors, it is now admitted that inflammation-driven recruitment from blood does not account for all ASCs present in the TME. TLSs can develop in chronic inflammatory sites and are organized similarly to SLOs, with B cell follicles adjacent to T cell-rich areas that contain mature DCs and high endothelial venules. They are associated with better patient prognoses in many solid cancers including BC and OC [54,55,56,57]. Numerous evidence support that TLSs are sites where B cells differentiate into ASCs in response to local presentation of tumor Ags. Indeed, in many epithelial tumors, virtually all B-cell differentiation stages, e.g., naïve B cells, activated B cells, germinal center (GC) B cells, memory B cells, PBs, and terminally differentiated ASC (PC), have been identified by flow cytometry and bulk or single transcriptome profiling [1,22,35,58], arguing for an ongoing local humoral immune response in certain tumors. In line with this, TLSs can harbor B cell follicles with GC containing B cells expressing Ki67 and AID, revealing ongoing Ag-driven expansion, somatic hypermutation (SHM), and class-switch recombination [1,49,51,59]. In addition, a strong associations between the presence of TLSs and elevated ASC numbers have been reported in various solid cancers [7].

Examination of the immunoglobulin repertoires of B cells and ASCs from TLS-positive tumors revealed a more oligoclonal response compared to tumors with an unstructured immune infiltrate [7,52,60,61]. Using spatial BCR profiling, a recent study in renal carcinoma revealed clonal selection and expansion in TLS areas and detected the presence of fully mature ASC clonotypes at distance from TLSs, indicating that SHM occurs in TLSs and that mutated ASCs disseminate throughout the tumors [51]. Our own study in ovarian tumors from patients with paraneoplastic cerebellar degeneration, a rare autoimmune neurological disease associated to cancer, revealed tumor Ag deposits in B cell follicles of TLSs [62]. TLSs may thus allow emergence of clonally expanded tumor Ag-specific ASCs. The latter have been recently detected in several cancers, including human papillomavirus (HPV) + head and neck [59] and ovarian [61] cancers. Mazor et al., showed that ASCs that infiltrate ovarian tumors were mutated, clonally expanded, and produced antibodies reacting against metalloproteinase (MMP) 14—an autoAg that is overexpressed in ovarian tumor cells—and able to bind to the surface of tumor cells [61]. Moreover, Meylan and collaborators observed higher proportions of IgG-labeled apoptotic tumor cells in TLS^+^ tumors compared to TLS^−^ tumors [51], indicating that TLS sustain the production of antibodies reacting with tumor cells and possibly leading to their elimination. 

Isotype switching to both IgGs and IgAs can occur in tumors [26,63,64]. It requires two signals: CD40L provided by T follicular helper (Tfh) cells and specific cytokines (produced by Tfh or surrounding cells), which dictate the nature of the switched isotype [65,66]. Three Tfh subsets can be distinguished according to their chemokine receptor and cytokine profiles, namely CXCR3^+^CCR6^−^ IFNγ-producing Tfh1, CXCR3^−^CCR6^−^ IL-4/IL-13-producing Tfh2, and CXCR3^−^CCR6^+^ IL-17/IL-22-producing Tfh17 cells [67]. In breast tumors, Tfhs are detected predominantly within fully developed TLSs, mostly consist of Tfh1 cells, and can trigger IgG and IgA production by B cells in vitro by a process involving CD40-CD40L interaction [68]. Therefore, it is more likely that the type of intratumorally produced cytokines largely dictates the class/subclass of antibodies expressed by newly differentiated ASCs. Karagiannis and colleagues showed in melanoma tissues that the presence of IgG4 ASC coincides with high levels of IL4 and IL10, two cytokines promoting Th2 polarization and isotype switching toward IgG4s [69,70,71]. Another study in a mouse model of prostate cancer demonstrated that TGF-βR signaling in B cells is mandatory for the induction of IgA PCs with immunosuppressive properties [9]. We can therefore reasonably suppose that the Ab isotype expressed by locally generated ASCs largely depends on the functional profile of Tfh cells and reflects the cytokine TME.

Altogether, our current knowledge indicates that Ti-ASCs may contain cells recruited from the circulation—possibly including some tumor Ag-specific cells generated in SLO—and cells that have differentiated locally in TLS from naïve and/or memory B cells and that are thus likely enriched in tumor Ag-specific cells (Figure 1). A recent study further supports this hypothesis by showing that tumors from patients with HPV-positive head and neck cancers were infiltrated by HPV-specific ASCs with minimal bystander recruitment of influenza-specific ASCs [59]. 

### 2.4. ASC and Antibodies Influence Cancer Patient Survival and Response to Immunotherapies

Numerous studies have evaluated the association between tumor infiltration by ASCs at the time of diagnosis and patient survival after surgery (Table 1). Overall, ASCs are usually associated with a better prognosis that is manifested by increased disease-free and/or overall patient survival. This positive correlation is reinforced in case of tumor co-infiltration by CD8^+^ T cells [7] and when tumor cells are coated with endogenous immunoglobulins, a situation that is usually accompanied by increased intra-epithelial T cells [35]. Nonetheless, a minority of studies reported an association with a poor prognosis [17,20,72,73]. These discordant results may be linked to the phenotypic and functional heterogeneity of the tumors ASCs infiltrate, as well as to methodological issues linked to the detection of ASCs, which range from a simple morphological identification, IHC/IF analysis with different markers like CD138 and IRF4, to computational analysis of ASC gene signature score from tumor bulk RNA-seq data using diverse algorithms. The use of CD138 as a marker of PC should indeed take into account that other cells, including epithelial cells, can express this cluster of differentiation and could bias prognosis studies [74]. Combining this marker with a morphological identification of PCs or with another ASC marker such as the Ig Kappa light chain (IGKC) should therefore be privileged [20,75,76]. Another possibility would be that the prognosis impact of ASC may vary with their state of maturation, as CD138 is upregulated during ASC maturation into PC but is poorly expressed at the PB stage [77]. The respective impact of PB versus PC in cancer remains so far largely unexplored.

Limited information is so far available regarding the impact of ASCs according to the Ab isotypes they produce. Most studies reported in the literature have analyzed the relation between the relative proportion in the TME of Ig subtypes and patient survival, assuming that Ig detected in the TME, or at the RNA or protein levels, are produced by Ti-B cells. Overall, in most cancers, including melanoma [63,78], lung cancer [79], bladder cancer [80], and prostate cancer [9], a high expression of IgG1 is associated with longer survival, whereas IgG4 and IgA expressions correlate with a negative outcome. However, IgAs were recently reported in OC as associated with a positive impact by impairing tumor growth [35]. In addition, in BC, it was shown that NY-ESO-1 more frequently elicited IgG response, which was associated with poorer prognosis, albeit IgG subclasses were not identified [50]. Moreover, it is difficult to interpret these data as NY-ESO-1, per se, is associated with shorter survival and this result may only highlight the cancer aggressiveness [81]. These results highlight the need to deeply characterize the diversity of ASC in the TME to better appreciate their impact on anti-tumor immunity and patient outcome.

Beyond their usual association with increased patient survival following surgery and chemotherapy, ASCs were recently shown to be also associated with a better response to the checkpoint blockade. By comparing the transcriptome of tumors from patients treated with nivolumab (anti-PD-1) +/− ipilimumab (anti-CTLA4), Helmink et al., documented significantly higher expression of ASC-related genes MZB1, JCHAIN, and Immunoglobulin Lambda Like polypeptide 5 (IGLL5) in patients that responded to treatment compared to nonresponsive patients [6]. Patil et al., showed that PCs, identified with a transcriptional signature including MZB1, DERL3, TNFRSF17, JSRP1, SLAMF7, and immunoglobulin genes, predicted a better overall survival to atezolizumab (anti-PD-L1) in nonsquamous cell lung cancer independently of intra-tumoral CD8^+^ T cells and PD-L1 expression [82]. In addition, Meylan et al., showed in clear cell renal cancers that patients with a high frequency of IgG-labeled tumor cells had a high response rate to nivolumab +/− ipilimumab and prolonged PFS [51]. The predictive impact of ASCs on immune checkpoint inhibitors may be explained in part by the activation of Tfhs, which strongly express PD1 leading to B cell activation and antitumor immune response, as suggested in murine models [83,84].

## 3. Antibodies Functions in Cancer

The canonical function of PCs is to produce antibodies. Depending on their isotype, isoform, and/or Ag target, they can differentially impact tumor cells and antitumor immunity and may thus account for the anti- or pro-tumor properties of tumor-infiltrating ASCs.

### 3.1. Antibody Isotypes and Isoforms Determine Their Functions

Antibody Fab regions are involved in recognition and possible modulation or neutralization of Ags on infectious particles or tumor cells, whereas antibody Fc regions can trigger a wide scope of immune effector functions, such as complement-dependent cytotoxicity (CDC), antibody-dependent cellular cytotoxicity (ADCC) or phagocytosis (ADCP) that can lead to tumor cell killing. It is well-documented that the latter functions contribute to the efficacy of several cancer therapeutic monoclonal antibodies (mAbs) such as Rituximab (anti-CD20) and trastuzumab (anti-HER2). Antibody isotypes and isoforms vary in their ability to activate or inhibit immune system components including the formation of the complement complex or the engagement of Fc receptors (FcRs) on the surface of effector cells [91]. This section will focus on antibody functions through their Fc region (Figure 2).

### 3.2. Antibody-Mediated Killing of Tumor Cells

Mouse models and the use of therapeutic monoclonal Abs (anti-HER2/neu, -EGFR) in humans have revealed that Abs specific to tumor Ags can kill tumor cells. Whether endogenously produced antibodies contribute to tumor cell control in cancer patients remains, however, to be formally demonstrated. Nonetheless, the fact that Abs from patients can exert cytotoxicity against tumor cells was established in 1979. Wood et al., indeed showed that the serums of patients with brain tumors, but not those of healthy volunteers, exerted significant cytotoxicity against allogeneic astrocytoma cells [92]. Later, other groups demonstrated that tumor-specific antibodies isolated from patients were effectively able to kill neoplastic cells in vitro and in vivo [93,94]. Recently, Meylan and collaborators found an association between IgG-stained malignant cells and apoptotic tumor cells in favor of an antitumor effector activity [51]. Such capacity to kill tumor cells varies strikingly depending on the Ab class and subclass.

IgM. Secreted IgMs harbor a pentameric form displaying 10 Ag-binding sites and thus constitute the isotype with the highest valency [95]. It is the first isotype produced after an initial immunological challenge. “Natural IgMs”—which are polyreactive, mostly recognize self-Ags with low affinity [96], and are present in the circulation even without Ag challenge [97]—seem to play a crucial role in the immunosurveillance of precancerous and cancerous cells [98]. Atif et al., indeed showed that mice lacking a diverse natural IgM repertoire display exacerbated tumor growth, and described a mechanism by which interstitial macrophages and monocytes cleared neoantigen-expressing cells coated with natural IgM antibodies [99]. Additionally, natural or adaptive IgM able to opsonize tumor Ags can directly kill malignant cells via the activation of the complement classical pathway [100].

IgG. There are four subclasses of IgG (IgG1, IgG2, IgG3, and IgG4) that differ by their Fc regions and capacities to exert a cytotoxic function. That response to mAb therapy is usually higher in patients with Fc receptor polymorphisms that increase affinity for IgGs (FcγRIIa-131H/R and FcγRIIIa-158V/F) supports the fact that the antitumor function of antibodies requires Fcγ receptor-expressing cells [101], for instance, NK cells, monocytes/macrophages, and/or neutrophils. IgG1 and IgG3 can induce ADCC by engaging FcγRIIIa on CD16+ NK cells [102] to trigger cell cytotoxicity through perforin and granzyme release [103]. Different studies proved that NKs are able to induce ADCC in tumors. In a model of HER2+/neu breast tumor xenograft, it was indeed shown that an anti-HER2/neu engineered antibody lacking the ability to bind FcγRIIIa was inefficient to prevent tumor growth, suggesting a potential role of NK in mAb efficacy [104]. Moreover, Hubert and collaborators observed formation of ADCC synapses in immunocompetent mice bearing syngeneic breast tumors treated with an antibody directed against Tn, a specific breast cancer Ag [105]. In humans, anti-HER2 IgG1 does not only suppress HER2 signaling in breast cancer cells but also mediates ADCC through NK cells, increasing the efficacy of HER2 therapy [106]. Moreover, monocytes and macrophages can mediate ADCP of IgG1-opsonized tumor cells. By engineering different IgG subclasses directed against a melanoma Ag, Karagiannis et al., showed that IgG1, but not IgG4, induced phagocytosis of tumor cells by monocytes in vitro and reduced tumor growth in vivo in a mouse model of subcutaneous human melanoma xenograft supplemented with human immune effector cells [69]. Additionally, by intravital microscopy in mice, Gül et al., were able to visualize phagocytosis of opsonized circulating tumor cells by macrophages resulting in the prevention of liver metastases [107]. That macrophages may induce tumor cell killing through ADCC in humans is suggested by the recent observation of Meylan and colleagues in renal cancer of the close proximity of macrophages and apoptotic tumor cells harboring IgG deposits [51]. Neutrophils are also efficient at performing cytotoxicity in an FcγRIIa (CD32a)-dependent manner [108]. Intriguingly, neutrophils use a mechanism different from those of NK cells or monocytes to kill tumor cells. In a model of HER2/neu+ breast cancer cells opsonized by anti-HER2 IgG1 Ab, Matlung et al., indeed demonstrated that this cytotoxicity was not dependent on lytic granule release and involved an active mechanic destruction of the target cell plasma membrane, leading to a form of immune cell-mediated necrotic type of cell death referred to as trogoptosis [109]. This method of killing is restricted to neutrophils, requires neutrophil/target cell conjugate formation dependent on CD11b/CD18 interaction, and is weakened by CD47-SIRPα interactions. Another important component that can mediate tumor cell killing, albeit frequently neglected in the TME, is the complement system. Complement cascades can be initiated by three different pathways: classical, alternative, and lectin pathways. Complement activation through the different pathways results in the generation of the terminal complement complex C5b-9, also called membrane attack complex (MAC), that elicits target cell lysis by creating transmembrane pores [110]. Deposition of MAC on tumor tissue has been described in different cancers, including breast and ovarian malignancies [111,112], suggesting that CDC is occurring in the TME. Cumulative evidence suggests that the classical pathway, which is mainly activated by IgM- or IgG-containing immune complexes (ICs), is implicated in complement cascade initiation in the TME [8,113]. Yet, CDC has been so far mainly demonstrated for therapeutic anticancer mAbs. Rituximab, an anti-CD20 mAb, was capable to induce activation of the classical complement pathway on malignant B cells in vitro and in vivo [114,115]. Nevertheless, tumor cells can increase the expression of inhibitory molecules of complement activation on their surface to escape CDC, as demonstrated in patients whose breast cancer cells overexpress CD55 and CD59, two molecules that negatively regulate the complement system. These patients displayed a higher rate of relapse after anti-HER2 treatment compared to patients with lower expression [116]. Inhibition of the expression of these molecules on breast cancer cell lines induces a significantly enhanced anti-HER2-induced CDC-dependent lysis [117].

Among IgG subclasses, IgG1 and IgG3 demonstrate the highest affinity for most FcγRs and are potent activators of ADCC, ADCP, and CDC. Such cytotoxicity can induce the release of danger-associated molecular patterns (DAMPs) and tumor Ags from necrotic tumor cells and may initiate durable adaptive antitumor immunity, including a cytotoxic T cell response. Gordan and colleagues revealed that a complex set of factors, including the organ environment, the level of Ag expression, and the antibody concentration, determines the activation of different FcγRs and/or the classical complement pathway resulting in the cytotoxic IgG activity [118].

IgA. A role of IgA in tumor immunosurveillance is suggested by the fact that individuals with IgA deficiency are at (moderate) increased risk of cancer, especially in the gastrointestinal tract (HR = 1.64) [119]. The functions of IgAs are highly diverse and depend on their subclasses, isoforms, and the nature of accessory cells expressing IgA receptors in the TME. Two IgA subclasses coexist, IgA1 and IgA2, that differ by the size of their hinge region, their shape, and their glycosylation profile [120]. In addition, IgAs can exhibit a monomeric form (mIgA) dominant in serum, a dimeric (dIgA) form containing the J-chain and characteristic of mucosal tissues, and a secretory form (SIgA) consisting of dIgA assembled to the secretory component extracted from the polymeric Ig receptor “PIgR” during epithelial IgA transcytosis [121]. Depending on their isoforms and subclasses, human IgAs can bind with different affinities to several specific receptors, including FcαRI/CD89 [122] and the polymeric Ig receptor (PIgR), as well as to accessory receptors like DECTIN-1, DC-SIGN, and CD71. FcαRI is expressed by monocytes, macrophages, Kupffer cells, eosinophils, and neutrophils [123,124]. Both mIgA_1-2_ and dIgA_1-2_ bind to FcαRI with moderate affinity, while IgA ICs bind avidly. By engaging FcαRI on neutrophils, IgAs can induce effective ADCC of various tumor targets, including HER2+/neu breast cancer cells [125,126,127]. Importantly, while neutrophils express high affinity receptors for both IgG and IgA, IgA appears more efficient than IgG to kill tumor cells, as shown with therapeutic antibodies in vitro and in vivo [126,127,128]. This may be due to a stronger ITAM signaling in neutrophils induced by IgA compared to IgG. Indeed, whereas FcγRIIa, the main FcγR expressed by neutrophils, can signal through only one ITAM, FcαRI can engage four ITAMs on the FcαRI-associated FcRγ-chains. As demonstrated for IgG, IgA induced neutrophil-dependent tumor cell killing likely involves a mechanism of trogocytosis that can be potentiated by the blockade of CD47-SIRPα interaction [129].

IgE. IgE plays a pathogenic role in allergies by triggering the rapid degranulation of FcεR-expressing cells including basophils and mast cells, and as a protective effector in parasitic infections, but its role in cancer remains poorly understood. Interestingly, omalizumab, a therapeutic mAb blocking IgE binding to its high-affinity receptor FcεR I and administrated to severe cases of asthma, has been associated with higher risk of solid cancers development, suggesting that endogenous IgE may be part of the immunological host defense against tumors [130]. Interestingly, Crawford and colleagues reported that topical exposure to an environmental DNA-damaging xenobiotic can induce an IgE response, depending on γδ intraepithelial T cells. This IgE response can protect against epithelial carcinogenesis by a mechanism involving engagement of FcεRI, possibly on basophils [131]. Moreover, Josephs et al., by engineering mAbs specific for folate receptors (FRα)—which are widely expressed by human ovarian tumor cells—demonstrated the superior antitumor efficacy of IgE compared to IgG in a syngeneic rat model of cancer [132]. IgE, but not IgG, promotes MCP-1 production by monocytes, resulting in increased recruitment of macrophages and their polarization towards M1-like type cells expressing CD80 and TNFα. Genes associated with the FcεR complex, including *FCER1* (α subunit of the IgE receptor) and *MS4A2* (β subunit of the IgE receptor), were associated with a favorable prognosis of patients with lung cancer [133]. These data support a tumor-protective role of IgE, but further studies are needed to determine whether an IgE response to tumor Ag develops in cancer patients and can be manipulated.

### 3.3. Antibodies Can Promote Anti-Tumor Immunity by Favoring Antigen Presentation to T Cells

Ag presentation is a critical step for inducing antitumor immunity and usually requires specialized Ag presenting cells (APC), notably for CD8^+^ T cells in the case of soluble or membrane bound Ags that require cross-presentation. Abs, able to form immune complexes (IC), change the way tumor Ags are captured and presented to T cells, and have been shown to be instrumental for inducing protective T cell-mediated immunity. 

When complexed to IgG, Ags can be internalized by APC, processed, and presented on major histocompatibility (MHC) class-II molecules to CD4^+^ T cells [134]. Importantly, IgGs can also deliver exogenous Ags into the MHC class-I processing pathway of DC for presentation to CD8^+^ T cells in a process known as Ag “cross-presentation”. Ag/IgG ICs have been shown to allow cross-priming of tumor-specific CD8^+^ T cells able to protect against the development of colorectal cancer and lung metastases, by a process involving the surface FcγR and cytoplasmic FcRn (neonatal Fc receptor for IgG) of DC. FcRn ligation by IgG-IC induces IL12 production by DC allowing CD8^+^ T cell polarization into IFNγ producing cytotoxic cells [135,136]. Such processes might be facilitated by TLS thanks to the presence of a contingent of mature DC in the vicinity of T cells and of ASCs producing antitumor Abs. 

It is possible that such capacity to favor Ag presentation is not limited to IgG. Indeed, certain DCs can use FcεRI for directing soluble Ags complexed to IgE into the cross-presentation pathway to prime anti-tumor CD8^+^ T cells able to eliminate cancer cells [137]. An in vivo study also recently revealed that natural IgMs allow monocytes to present tumor neoantigens to CD4^+^ T cells leading to the emergence of activated T helper cells expressing CD40L with the capacity to license Ag-cross-presenting Batf3+ conventional type 1 DC for induction of a cytotoxic CD8^+^ T cell response [99]. Although FcαRI is expressed on immature DC and can internalize IgA1 or IgA2 complexes, Ag presentation to T cells appears rather inefficient through this pathway [138]. Yet, engagement of IgA receptors on inflammatory-type DC may indirectly enhance Ag-specific CD8^+^ T cells responses through its capacity to activate DC as recently shown for IgA2 [139]. 

Thus, antibodies may play an essential role for bridging innate and adaptive immunity to initiate or amplify efficient antitumor immune responses contributing in fine to tumor cell elimination. 

### 3.4. IgA Transcytosis in Tumor Cells May Increase Sensitivity to CD8+ T Lymphocytes Cytotoxicity

Early studies have revealed the existence of IgG and/or IgA deposits in/on tumor cells in certain tumors, including breast and ovarian primary tumors [35,80]. The team of Conejo-Garcia recently provided insights into the mechanism underlying IgA penetration in tumor cells and its consequences. Using tumor cell lines, they showed that polyclonal IgA can be internalized by ovarian and endometrial tumor cells after binding to the polymeric IgA receptor (PIgR) resulting in drastic transcriptional changes in malignant cells [35,140]. In ovarian tumor cells, these changes included inhibition of the RAS pathway and upregulation of inflammatory genes, and were associated with an increased sensitivity of tumor cells to cytolytic killing by T cells in vitro and in vivo [35]. While both irrelevant and tumor-specific IgA can mediate these effects, they were more pronounced when tumor cells expressed the cognate Ag. Whether a similar mechanism exists for IgG is currently unknown; in breast tumors, FcRn, an essential IgG transporter, can be expressed in malignant epithelial cells of mammary glands, but IgG internalization through this receptor have never been demonstrated [141].

### 3.5. Antibodies Can Suppress Tumor Cell Killing

IgG4 is the less abundant subclass of IgG in the serum and has limited effector functions due to its negligible binding ability to C1q and Fcγ receptors compared to other IgG subclasses. There is evidence that this IgG subclass dampens antitumor immunity. Karagiannis et al., indeed reported that metastatic melanoma patients with a higher serum IgG4/total IgG ratio had significantly lower survival rates and that tumor-specific IgG4s were not only unable to activate monocytes to kill tumor cells but also suppressed tumor-specific IgG1-induced ADCP of tumor cells, indicating that IgG4s can suppress tumor-cell killing likely by competition with IgG1 for FcγRI binding [69]. Non-tumor-specific IgG4s also exerted a partial suppressive function. In addition, Alberse et al., reported that IgG4s can interact with IgG1 and IgE through their Fc domains, potentially influencing their antitumor effector functions [142]. Besides the identity of the Ab subclass, several post-translational modifications can drastically affect the antitumor function of IgG. A substantial fraction of IgG in cancer patients are indeed sialylated in their Fc part, conferring to these Abs a potential suppressive function [143]. Zhang et al., provided evidence that the cleavage of single-peptide bonds in the hinge region of endogenous IgG1 (sc-IgG1) by breast cancer-associated MMP constitutes an evasion mechanism to humoral immunity in cancer. ScIgG1s are significantly higher in tumors than in normal tissues, positively correlate with adverse clinical factors, such as frequent local metastasis to axillary lymph nodes, and display reduced Fc immune effector functions, including ADCC [144]. Such IgG alterations have also been observed for an anti-HER2 mAb infused in patients with BC leading to a reduction of its therapeutic efficacy [145]. How the TME affects the functions of the different Ab isotypes and the consequences of these modifications on tumor immune evasion needs to be analyzed.

IgA. Polymeric IgAs have been involved in several inflammatory diseases, including IgA nephropathy, since their aggregation strongly increases their affinity to FcαR1 allowing potent cell activation and initiation of an inflammatory response. Conversely, serum mIgA in steady state are endowed with powerful anti-inflammatory properties toward the immune system. IgA poorly activate the complement pathway and have been shown to inhibit NK-dependent ADCC [146]. Moreover, in the absence of bound Ag, mIgA impair IgG-mediated phagocytosis, chemotaxis, bactericidal activity, oxidative burst activity, and cytokine release [147]. Pasquier and colleagues further demonstrated that mIgA binding to FcαRI inhibits IgG-mediated phagocytosis in human monocytes, by a process involving intracellular signaling through the associated FcRγ adaptor that contains an inhibitory ITAM configuration [148]. Eventually, once bound to FcαRI, these Abs may induce death of activated neutrophils instead of tumor cell trogoptosis [149]. Therefore, the dual role of IgAs largely depends on their isoforms, the presence or absence of target Ags, and the nature of accessory cells that they can engage.

### 3.6. Antigen/Antibody Immune Complexes Tumor-Promoting Inflammation

Tumor-promoting inflammation is the seventh hallmark of cancer and contributes to various aspects of solid tumor development, including growth and survival of cancer cells [150]. Numerous studies highlighted that IgG-containing ICs can trigger inflammation in the TME, which in turn can promote carcinoma development. Indeed, high levels of circulating ICs are associated with increased tumor burden and poor prognosis in various cancers, including BC [151]. Using a transgenic mouse model of epithelial carcinogenesis, Andreu and colleagues showed a reduction of both leukocyte recruitment and tumor growth in the absence of B cells or activating FcγR. Such experiments revealed that IgG-mediated stimulation of mast cells and macrophages can promote, in certain settings, inflammation and a subsequent increase of angiogenesis, survival, and proliferation of tumor cells [152]. Furthermore, Ags opsonized by IgGs can activate the classical complement pathway leading to the release of the C3a and C5a anaphylatoxins, whose levels correlate with decreased effector T cell numbers and increased tumor burden in a mouse model of OC [153]. C3a and/or C5a can indeed directly enhance tumor cell survival, proliferation, and metastasis after binding to their receptors on tumor cells. Moreover, they can recruit myeloid-derived suppressor cells and tumor-promoting macrophages, stimulate the protumorigenic properties of mast cells resulting in chronic inflammation, suppress CD8^+^ T cell cytotoxicity, and promote formation of protumoral neutrophil extracellular traps [8].

### 3.7. Modulation of Tumor-Associated Antigens by Antibodies

Antibodies may also affect tumor cells, independently of the immune system, by directly exerting an agonist or antagonist function on target Ags, such as growth factors or surface receptors that are essential for tumor cell survival, expansion, or dissemination, or by directly triggering tumor cell apoptosis upon binding to cell surface proteins. These properties are largely exploited for cancer therapy with the success of mAbs targeting HER2, EGFR, and other growth factor receptors. There is some evidence that such Abs can naturally be generated in cancer patients. For example, natural IgMs are able to induce tumor-specific apoptosis by cross-linking of complement inhibitory proteins, blocking growth factor receptors or by increasing the intracellular level of neutral lipids [98,154,155]. In addition, a monoclonal antibody directed against the 78 kDa glucose-regulated protein (GRP78)—an Ag highly expressed by breast tumor cells [156]—and isolated from a gastric cancer patient [155] was shown to be able to directly kill cancer cells by cellular lipotoxicity involving intracellular lipids, cholesterylester, and triglycerides accumulation and leading to cell apoptosis. Another example relies on the pathologic effects of Abs that can be observed in cancer patients with a paraneoplastic syndrome. In these rare autoimmune diseases, high levels of auto-Ab directed against self-Ags aberrantly expressed in tumor cells are usually observed. In the case of ovarian tumors with Abs to N-methyl-D-aspartate receptor (NMDAR) [157,158], incubation of the Ab on neuronal cells has been shown to result in the rapid internalization of the extracellular GluN1 NMDAR subunit and decrease of glutamatergic synaptic function, possibly explaining neurological clinical manifestations observed in patients [159,160]. A minor fraction of patients affected by lung or breast cancer, ovarian teratoma or thymoma can develop a paraneoplastic syndrome caused by the humoral and cellular immune response against GABA-B receptor, expressed by both neurons and tumor cells. Anti-GABA-B autoantibodies found in patients’ serums were shown to colocalize with GABA-B receptor and to disrupt its structure resulting in seizures and changes in memory, learning, and behavior in animal models [161,162].

Overall, antibodies can have both beneficial and deleterious roles in the TME, and their better characterization and the understanding of their mode of action will be essential for manipulating them for therapeutic purposes. 

## 4. Antibody-Independent Pro- vs. Antitumor Functions of Antibody-Producing Cells

Besides the production of antibodies, there is increasing evidence that ASCs display several noncanonical functions that can impact tumor growth and antitumor immunity.

### 4.1. Cytokine Secretion

In addition to antibody production, accumulating evidence demonstrates that certain human ASC subsets can produce a diverse array of cytokines or molecules endowed with either pro- or anti-inflammatory functions [163]. Nevertheless, the cytokine expression profile of Ti-ASC remains so far poorly documented, possibly due to their usual low frequency and to the fact that detecting intracellular cytokines in B cells can be technically challenging. Inflammatory bowel disease associated-ASC can produce granzyme B [164,165], traditionally known for its perforin-dependent proapoptotic function underlying the capability of cytotoxic immune cells to kill tumorigenic cells [166,167]. Moreover, in infectious disease, ASCs can express the inducible Nitric Oxid Synthase (iNOS) [168], which was reported in BC to elicit cancer progression through regulation of cell adhesion and motility [169,170]. ASCs can also secrete immunosuppressive molecules such as IL-35 [171] that inhibit antitumor cytokines including IFN-γ [172]. Interestingly, PCs in human lung cancers can express EBI3, a component of IL-27 and IL-35 heterodimers [173], and IL-10 [9,10,11] a cytokine with both tumor-promoting properties through inhibition of tumor Ag presentation [12,13] and tumor-inhibiting activities through anti-angiogenic functions via down-regulation of VEGF, IL-1b, TNF-α, IL-6, and MMP-9 [174]). IL-10-expressing PB have been described in human renal cell carcinoma and appeared to be enriched in IgA producing cells [14]. IL-10^+^ PB infiltration was associated with an upregulation of the T cell exhaustion marker Tim-3 and higher tumor size and stage, suggesting that they may suppress T cell immunity. Using mouse models of prostate and liver cancers, Shalapour and colleagues showed that IL10-producing IgA^+^ secreting cells are induced in the TME, especially after chemotherapy, by a process requiring TGFβ-receptor signaling in B cells, and can directly suppress cytotoxic CD8^+^ T lymphocytes in an IL-10 dependent manner [9,10]. Thus far, little is known about the specific markers that could distinguish suppressive IL-10^+^ ASC from their non-IL-10-producing (potentially antitumor) counterparts. Regulatory IL-10^+^ ASCs producing IgM or IgG isotypes have also been described in autoimmune and infectious diseases [9,10,15,16].

### 4.2. Other Functions of ASCs in the TME

IgA^+^ ASCs in prostate and liver cancers also express high levels of PD-L1 on their surfaces [9,10]. PD-L1 neutralization did not decrease tumor burden in mice models deficient in IgA or CD8 T cells, indicating that it acts via IgA^+^ cells and targets CD8^+^ T cells. PD-L1 expression on ASC was confirmed in human samples from BC [175]. In infectious disease, the Fillatreau group recently described LAG3 as another immune checkpoint expressed by regulatory ASCs [176]. These cells were a major source of IL-10 early after infection with *Salmonella* [176]. They also express the PD-L1, PD-L2, and CD200 (Lino et al., 2018) inhibitory receptors that are upregulated upon activation and frequently associated with T cell. Regulatory ASCs, possibly present in tumors, may use these inhibitory receptors as additional mechanisms to suppress an immune response in addition to secretion of IL-10 or other suppressive cytokines.

The increased expression of immune checkpoint ligands on ASCs may suggest close interactions between ASCs and other immune cells in the TME. Recent evidence also suggests a role for exosome transfer in the function of Ti-ASCs. In OC, it has been shown that ASCs were enriched in the mesenchymal subtype of HGSOC and that the epithelial to mesenchymal phenotypic switch of tumor cells in vitro and in vivo, which controls tumor metastasis and patient prognosis, was mediated by the transfer of PC–derived exosomes containing miR-330-3p into non-mesenchymal ovarian cancer cells inducing their transition [73] (Figure 3).

## 5. Conclusions

In this review, we have emphasized that ASCs can exhibit both beneficial and detrimental roles in the TME. These pro- and antitumor functions depend not only on the class and subclass of Abs they produce, the identity of their target Ag, and the nature and functions of the accessory cells present in the TME, but also on their noncanonical activities such as cytokine production and immune checkpoint expression, the latter remaining to be better characterized. ASCs thus emerge as critical actors of the TME, but many questions remain to be addressed to envisage their therapeutic manipulation. To what extent is the ASC infiltrate fed by cells differentiated in secondary lymphoid organs vs. locally in TLS, and, in the latter case, do all ASCs originate from a germinal center reaction or can they also be generated in extrafollicular areas? Does the origin of ASCs impact their functional properties and Ag specificity? Although the diversity of Ti-ASCs is beginning to be revealed, a more comprehensive analysis of the Ab classes (including IgE) and subclasses, isoforms, and glycosylation patterns is required to better appreciate how ASCs may impact tumor immunity through Ab production. A particular attention should be paid to the contribution of the different IgA isoforms to activation versus immune tolerance. In addition, because Ab-related ASC functions largely depend on engagement of Ig Fc receptors on accessory cells, which identity will dictate Ab effector function, a comprehensive spatial and functional characterization of cells (neutrophils, NK cells, mast cells, macrophages) expressing the different Fc receptors should be performed to better elucidate/predict the function of ASCs. Furthermore, the presence of Ti-ASCs and TLS has recently been associated with better response to immune checkpoint inhibitors. Determining whether ASCs and Abs play active roles in this response and identification of the underlying mechanism could pave the way for defining strategies to improve the efficacy of immunotherapies. Increasing recruitment, neodifferentiation, and/or function of antitumor ASCs could indeed complete the anticancer therapeutic arsenal. Inducing TLS could be an option, since these structures favor emergence of ASCs enriched in tumor Ag-specific cells and production of antibodies able to bind to tumor cells. This will require a better understanding of how cancer-associated TLS are initiated and maintained in an active form. Finally, although evidence of a detrimental effect of ASCs are still scarce in humans, one therapeutic avenue in certain cancers could consist of inducing their specific depletion. This could be achieved through proteasome inhibition with the risk of also depleting antitumor ASCs. Another important challenge will thus consist in identifying reliable markers able to distinguish pro- from antitumoral ASCs and the signals involved in their development in order to identify the best therapeutic strategies.

## Figures and Tables

**Figure 1 cancers-14-04800-f001:**
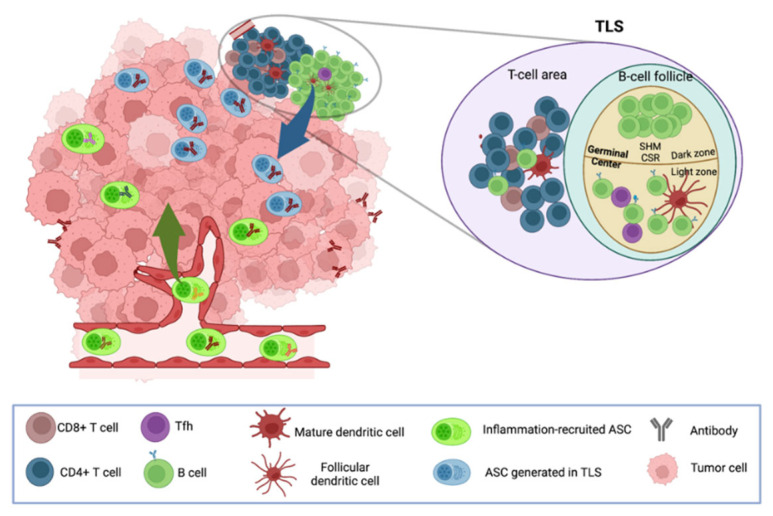
Possible origins of tumor-infiltrating ASCs. Circulating polyclonal ASCs can be recruited to the tumor site in response to inflammation, whereas oligoclonal ASCs can be locally generated from naïve and/or memory B cells in TLS in response to stimulation by Ags, including tumor Ags.

**Figure 2 cancers-14-04800-f002:**
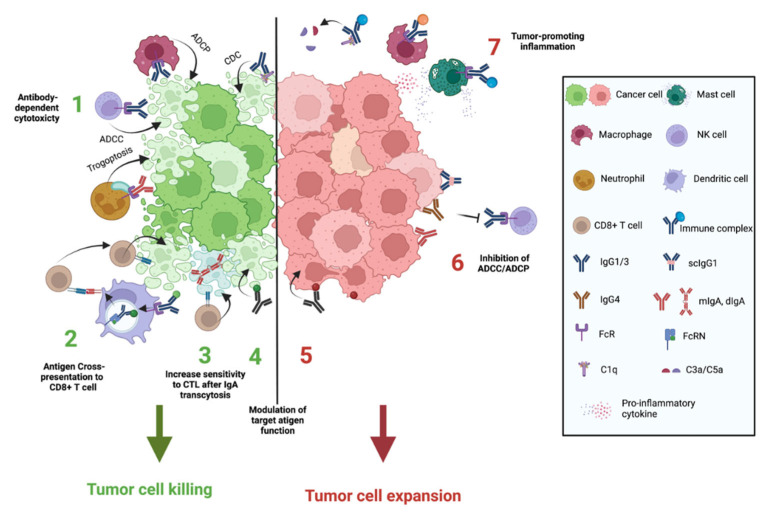
Mechanisms of action of antibodies in cancer. Antibodies can use several mechanisms to suppress (1–4) or promote (5–7) tumor growth. (1) Certain subclasses of antibodies can mediate tumor cell killing by ADCC mediated by NK cells, ADCP by macrophages, trogoptosis by neutrophils, and CDC. (2) Capture of antibody/antigen complex by DC can prime anti-tumor immunity through antigen cross-presentation to CD8+ T cells. (3) Transcytosis of dimeric IgA in tumor cells increases their sensitivity to CD8+ T cell-mediated cytotoxicity. (4) Direct modulation of target antigen function leading to tumor cell dysfunction/elimination. (5) Modulation of tumor antigens leading to survival/proliferation of tumor cells. (6) IgG4, scIgG1, and IgA can suppress the effector functions of cytotoxic antibody isotypes. IgG4 compete with IgG1 for FcγR binding. ScIgG1 has a reduced ability to induce ADCC compared to IgG1. IgA inhibits NK-dependent ADCC mediated by IgG1/IgG3. (7) IgG-antigen immune complexes can induce tumor-promoting inflammation by stimulation of macrophages and mast cells and/or by the activation of the classical complement pathway and the release of pro-inflammatory anaphylatoxins.

**Figure 3 cancers-14-04800-f003:**
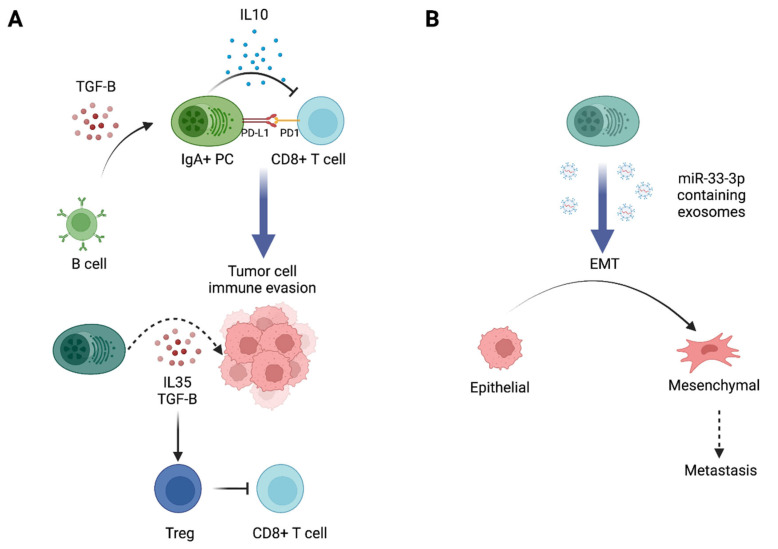
Noncanonical functions of antibody-secreting cells involved in tumor promotion. (**A**) IgA+ PC can secrete IL-10 and express the inhibitory immune check-point ligand “PD-L1” and suppress CD8+ T cell function. PC may secrete other suppressive cytokines, like IL-35 and TGFb, involved in tumor promotion by stimulating tumor cell proliferation and/or suppression of antitumor immunity. (**B**) PCs are able to deliver mIR-330-3p containing exosomes to tumor cells initiating an epithelial to mesenchymal phenotypic switch and tumor dissemination.

**Table 1 cancers-14-04800-t001:** ASC impact on patient prognosis. Studies reporting a positive impact are highlighted in blue, while those documenting a negative impact are highlighted in salmon.

Author/Year	Histological Tumor Type	Number of Patients	Identification of ASC	Prognosis	Reference
**Kroeger et al., 2016**	HGSOC	30	CD20^−^CD38^+^CD138^+^cytosolicCD79a^+^ IHC CD19^+^IgD^−^CD38^+^ Flow Cytometry *TNFRSF17/IGJ* PC gene signature	Good	[7]
**Lundgren et al., 2016**	OC	209	CD138 IHC *IGKC* gene expression	Poor Neutral	[20]
**Yang et al., 2021**	HGSOC	351	Gene signature (CIBERSORT)	Poor	[73]
**Biswas et al., 2021**	HGSOC	534	CD19^+^CD138^+^ (multiplex IHC) Internalized IgA in tumor cells	Good (total area and epithelial tumor islets) Good	[35]
**Schmidt et al., 2012**	BC OC	1810 426	*IGKC* expression	Good Neutral	[76]
**Iglesia et al., 2014**	BC OC	728 266	IgG cluster	Good (nonluminal BC) Good (mesenchymal and immunoreactive molecular subtypes)	[85]
**Gentles et al., 2015**	Pan-cancer	796 BC 1127 OC	Plasma cell gene signature (Cibersort)	Good (BC) Neutral (OC)	[86]
**Ridolfi et al., 1977**	Infiltrating ductal carcinoma	192	Morphological identification on hematoxylin and eosin-stained slides	Neutral (Medullary carcinoma) Good (others)	[87]
**Yeong et al., 2018**	Triple-negative BC	269	intratumoral CD38^+^ IHC stromal CD38^+^ IHC	Good Neutral	[25]
**Mohammed et al., 2012**	Invasive ductal breast cancer	468	CD138^+^ IHC and morphological identification (H&E)	Poor	[17]
**Miligy et al., 2017**	Invasive BC	44	CD138^+^ IHC	Neutral	[23]
**Kuroda et al., 2021**	TNBC	114	Stromal CD38^+^ IHC Intratumoral CD38^+^ IHC, stromal and intratumoral CD138^+^ IHC	Good Neutral	[88]
**Fan et al., 2011**	BC	550	IGG gene cluster expression	Good	[89]
**Harris et al., 2021**	TNBC	69	Plasma cell signature (Cibersort)	Neutral	[90]
**Wei et al., 2016**	BC	92	Morphological identification (typical “cartwheel” nucleus)	Poor	[72]

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
