# Peer review of "Heterogeneity and Functions of Tumor-Infiltrating Antibody Secreting Cells: Lessons from Breast, Ovarian, and Other Solid Cancers"

_cancers, 2022, doi:10.3390/cancers14194800_

Round 1

Reviewer 1 Report

The authors have amassed a timely review to show the clinical significance of the tumor-infiltrating B cells (TiBs) and antibody-secreting cells (ASC). They have discussed the several mechanistic underpinnings of these cells and their impact on anti-tumor immunity. Moreover, they have also included plausible strategies to modulate these key entities to improve cancer therapeutics.

Comments:

1. The authors may discuss the several subtypes of B-cells and how they modulate the immune microenvironment mainly associated with the ASCs. A figure may help as well.

2. The authors may enlighten the readers with the biological tools that may be used to study the TiBs and ASCs in the tumor microenvironment.

2. The authors may include a “Simple Summary” section before “Abstract” for the broader impact of the article.

3. Several changes in the font style of the text may be addressed.

Author Response

We thank the reviewer for his/her comments and suggestions to improve the manuscript. A revised manuscript with revisions highlighted in yellow is provided.

Reviewer#1

1- The authors may discuss the several subtypes of B-cells and how they modulate the immune microenvironment mainly associated with the ASCs. A figure may help as well.

Authors’ response:

The different subtypes of B cells are no briefly discussed in the introduction.

2- The authors may enlighten the readers with the biological tools that may be used to study the TiBs and ASCs in the tumor microenvironment.

Authors’ response:

The various tools to study Ti B cells are now mentioned in the introduction. The various markers that can be used to study ASC were already mentioned in the original version of the manuscript.

3- The authors may include a “Simple Summary” section before “Abstract” for the broader impact of the article.

Authors’ response:

A simple summary has been added as requested.

4- Several changes in the font style of the text may be addressed.

Authors’ response:

Changes in the font styles have been made to facilitate reading.

Reviewer 2 Report

B cells and antibody secreting cells (ASC), which have emerged as key participants in the tumor microenvironment, play a vital role in influencing the anticancer immune response. Indeed, ASC play a major role in tumorigenesis, patient survival, and treatment response. In this article, the authors reviewed the heterogeneity, and functions of antibody secreting cells in breast and ovarian cancers. They produced several positive outcomes from the study. I have few minor minor issues and suggestion with this article.

I suggest the authors modify the title because it doesn't seem like the article is specifically on breast and ovarian cancer from reading it.

Although the authors focused on the impact on solid tumors, I advise considering other types of tumors as well, including liquid tumors in a different category.

I suggest including information on these tumors' clinical trials.

A few references are missing for example, Cytokine secretion “In addition to antibody production, accumulating evidence……anti- inflammatory functions”.

Author Response

We thank the reviewer for his/her comments and suggestions to improve the manuscript. A revised manuscript with revisions highlighted in yellow is provided.

Reviewer#2

1- I suggest the authors modify the title because it doesn't seem like the article is specifically on breast and ovarian cancer from reading it.

Authors’ response:

We have modified the title as follows “Heterogeneity and functions of tumor-infiltrating antibody secreting cells: lessons from breast, ovarian and other solid cancers”.

2- Although the authors focused on the impact on solid tumors, I advise considering other types of tumors as well, including liquid tumors in a different category.

Authors’ response:

As our review aims to give an overview of the diversity and functions of non-malignant antibody secreting cells that infiltrate tumors, we have deliberately focused on solid cancers. This is now clearly mentioned in the title.

3- I suggest including information on these tumors' clinical trials.

Authors’ response:

See former point.

4- A few references are missing for example, Cytokine secretion “In addition to antibody production, accumulating evidence……anti- inflammatory functions”.

Authors’ response:

We have added several references throughout the manuscript when they were missing.